# All-MEMS Lidar Using Hybrid Optical Architecture with Digital Micromirror Devices and a 2D-MEMS Mirror

**DOI:** 10.3390/mi13091444

**Published:** 2022-09-01

**Authors:** Eunmo Kang, Heejoo Choi, Brandon Hellman, Joshua Rodriguez, Braden Smith, Xianyue Deng, Parker Liu, Ted Liang-Tai Lee, Eric Evans, Yifan Hong, Jiafan Guan, Chuan Luo, Yuzuru Takashima

**Affiliations:** James C. Wyant College of Optical Science, University of Arizona, Tucson, AZ 85719, USA

**Keywords:** lidar, MEMS, digital micromirror device, MEMS mirror, time-of-flight, solid-state lidar

## Abstract

In a lidar system, replacing moving components with solid-state devices is highly anticipated to make a reliable and compact lidar system, provided that a substantially large beam area with a large angular extent as well as high angular resolution is assured for the lidar transmitter and receiver. A new quasi-solid-state lidar optical architecture employs a transmitter with a two-dimensional MEMS mirror for fine beam steering at a fraction of the degree of the angular resolution and is combined with a digital micromirror device for wide FOV scanning over 37 degree while sustaining a large aperture area of 140 mm squared. In the receiver, a second digital micromirror device is synchronized to the transmitter DMD, which enables a large FOV receiver. An angular resolution of 0.57°(H) by 0.23° (V) was achieved with 0.588 fps for scanning 1344 points within the field of view.

## 1. Introduction

Laser beam steering (LBS) is one of the critical building blocks for optical instruments and applications ranging from industrial sectors, such as object tracking and optical interconnects, to scientific research tools, such as optical tweezers [1,2,3,4,5]. Additionally, in the consumer sector, LBS research has especially gained attention for lidar for autonomous and safe driving systems. Lidar employs range finding using methods such as direct time-of-flight (ToF), frequency modulated continuous wave (FMCW), intensity-modulated continuous wave IMCW), and time gating as a variation of ToF lidar [6]. Since automotive lidar involves both imaging and range finding, performance metrics, such as the maximum detectable range, angular throw or scanning field of view (FOV), angular and distance resolution, beam scan speed, random access capability, and mechanical packaging volume, are of interest, and these performance metrics often conflict with each other. Among them, the beam area (*A*) of the laser transmitter for LBS and the receiver is particularly critical because of the maximum range scales of $\sqrt{A}$ [6].

For long-range lidar to accommodate a large beam area, often mechanical scanning, such as rotating mirrors and polygon scanners, are employed. This approach requires large and heavy mechanical moving parts, and thus, the scan speed decreases as the beam area increases, and/or for a large device volume, higher power consumption is consequently required.

Non-mechanical LBS modalities, such as diffractive beam steering using liquid crystal on silicon (LCoS) spatial light modulators (SLMs), have been adopted for high-efficiency beam steering [3,4]. However, the FOV is limited to several degrees due to the large pixel period of LCoS SLMs compared to the wavelength. Moreover, the slow response time of liquid crystals, in milliseconds, limits the speed of scanning to be sub-kHz. Additionally, linearly polarized light is required for LCoS SLMs that reduce the photon throughput by half for a high-power unpolarized laser source. Such a drawback of an LC-based device can be overcome by a large-scale optical phased array [7,8]. However, the beam size and modulation speed are still behind the requirements, and they are still elusive for commercial products particularly.

In between the bulky and completely mechanical and non-mechanical scanning modalities, micro electro mechanical system (MEMS)-based LBSs are uniquely positioned. For example, a resonant MEMS’s mirrors are reliable and benefit high-speed operations up to tens of kilohertz while accommodating tens of degrees of FOV, although there is a trade-off between the FOV and the beam size. In general, a large angular throw requires low inertia or a correspondingly small mirror size, or vice versa [9].

Regarding another class of MEMS devices, an MEMS mirror array, such as a digital micromirror device (DMD) or an MEMS phase light modulator (PLM), is uniquely positioned, especially in terms of their large device area (i.e., $140\;{\mathrm{mm}}^{2}$), high-speed operation in several tens of kilohertz, and stable operation under harsh temperatures, i.e., an array temperature of 40 to 105 °C [10,11,12,13]. In particular, with a pulsed illumination, which is synchronized to the micromirrors’ movements, it is reported that DMDs efficiently steer the beam to a maximum angler throw of 48 degrees [14,15]. For the time-of-flight (ToF) lidar transmitter and receiver, a DMD’s large device area, high-speed operation, high efficiency, and large angular throw are attractive, particularly for maximizing transmitted energy while not exceeding eye-safe operations [14]. However, the drawback of this approach is the limited number of scanning points, up to several [15]. The challenge of the limited number of scanning points was addressed by increasing the number of pulses per mirror transition as well as the number of laser sources, though the improvement is still an order of magnitude [16].

To implement a lidar transmitter and receiver by using MEMS scanning devices, we report on an all-MEMS lidar system employing two kinds of MEMS—a DMD and a two-dimensional (2D) MEMS mirror for the transmitter, and a second DMD with light-pipe-based optics for the receiver. For the transmitter, fine and large-angle steering was employed with a 2D-MEMS mirror, but the beam area was rather small, in millimeters. The fine steering pattern is relayed to the DMD while matching the etendue (product of the area and angular throw) to that of the DMD by relay optics. In the optical architecture employing DMD and 2D-MEMS for transmitter, the DMD sequentially steers the fine steering pattern formed by the 2D-MEMS mirror into seven diffraction orders at 905 nm to achieve a total FOV of 38(*H*) × 7(*V*) degrees. For the receiver, a second DMD operates in a synchronous manner to the first DMD of the transmitter so that the returning signal from the object is redirected into the light pipe attached to the avalanche photo diode. The synchronous operation of the two DMDs and the 2D-MEMS mirror enables the all-MEMS and wide FOV beam steering for the transmitter and the wide FOV receiver in a quasi-solid-state implementation.

In this paper, we review the optical architecture of the hybrid MEMS lidar in Section 2 and address the optical design of the hybrid MEMS lidar transmitter and wide FOV receiver along with the control electronics to obtain lidar images. Section 3 addresses the experimental results, including the diffraction efficiency of the DMD transmitter and receiver, the distance and angular accuracy and their range, as well as still and live lidar image captures. Section 4 discusses the scalability and limitation of the proposed approach and the future direction of the proposed hybrid MEMS lidar.

## 2. Dual MEMS Lidar Architecture

### 2.1. Lidar Optical Architecture with DMD and MEMS Mirror

The lidar radiometric equation shows the strength of the return signal from the target scales with the beam area of the receiver (Rx). In addition, a larger beam area for the transmitter (Tx) is beneficial for decreasing the power density while ensuring eye safety [6]. In this hybrid MEMS lidar architecture, a Texas Instruments digital micromirror device (DMD) was used as the front end of the Tx and Rx to take advantage of its large device area of $140\;{\mathrm{mm}}^{2}$ [10]. A DMD is an arrayed MEMS mirror device that spatially modulates light by tilting mirrors between two binary states—the on-state and off-state, corresponding to the tilt angle of the micromirror at ±12 degrees. The low inertia of the micromirrors enables fast switching between the two states from several tens to 100 kHz [10]. When the DMD is used in the traditional binary (on- and off-state) amplitude modulation mode, the laser beam is steered only in two directions, +24 and −24 degrees, with respect to the normal of the DMD. The limited number of steering angles prohibits DMDs from being used for high-efficiency lidar Tx and Rx. It is also known that the limited number of steering points can be overcome by holographic beam steering by displaying a computer-generated hologram (CGH) in the lidar application [17]. However, an amplitude-modulated CGH suffers from a low diffraction efficiency—up to 10% in theory—and the angular extent is limited to the order of λ/p, where λ is the wavelength and p is the pixel period of the DMD [18]. Using a DMD for binary phase modulation is proposed to increase the diffraction efficiency to 40.5% in theory [19]; still, the maximum angular throw is limited to λ/p.

Recently, discrete beam steering using a DMD with a high diffraction efficiency was demonstrated and applied for a time-of-flight (ToF) lidar [15]. In a DMD, between the on- and off-states, there is a transitional period where the tilt angle of the micromirrors continuously changes from −12 to +12 degrees. The time scale of the transitional period is microseconds, whereas the pulse length for the laser pulse used for the ToF lidar is nanoseconds. By synchronizing the nanosecond-pulsed laser to the dynamic tilt movement of the micromirrors, it is possible to satisfy the blazed condition that diffracts light with high efficiency [15]. In this manner, the beam is efficiently steered into 7 and 10 diffraction orders, for example, at a wavelength of λ = 905 nm and 532 nm, respectively, with an angular throw of ±24 degrees. At λ = 905 nm, which is typically used for a ToF lidar, each of the diffraction orders accommodates about 5~6 degrees of horizontal sub-FOVs. The basic idea of the proposed two-dimensional MEMS mirror (2D-MEMS) and DMD hybrid optical architecture is that a small FOV, but fine scanning pattern of the 2D-MEMS mirror is diffractively and efficiently duplicated over the diffraction orders of the DMD, which enables fine and wide FOV beam steering. 

Figure 1a shows the optical architecture of the lidar transmitter (Tx) and receiver (Rx). In the Tx, the 2D-MEMS mirror scanned the sub-FOV (Figure 1b) in a raster scanning manner. The extent of the horizontal sub-FOV was matched to the angular separation of the diffraction orders of the DMD, at 5~6 degrees at λ = 905 nm with the DMD (DLP7000, Texas Instruments, Dallas, TX, USA). The two-dimensional fine steering pattern was sequentially diffracted to each of the diffraction orders of the DMD. Figure 1b,c shows the scan sequence. Since the DMD operated faster than the 2D-MEMS mirror, the scan points were sequentially steered across the diffraction orders for each of mirror tilt positions of the 2D-MEMS, as depicted in Figure 1c. In Figure 1c, we denote the position of the spot (i, j), where i and j are the position of the spot within a sub-FOV (1<i<N), and over diffraction orders (−3<j<3), respectively. First, the 2D-MEMS mirror was set to the initial position of (i,j)=(1,−3). Next, the DMD diffractively steered the beam into seven diffraction orders in such a way that the position of the beam shifted as (−3, 1)→(−2, 1)→(−1, 1)→(0, 1)→(+1 1)→(+2, 1)→(+3, 1). The scan sequence was repeated by increasing the value of the beam index i until the entire sub-FOV was scanned by the 2D-MEMS mirror.

While the Tx operated as explained, the receiver (Rx) DMD was synchronized to the transmitter (Tx) DMD so that the returning signal from an object from each of the diffraction orders was diffracted towards the single receiver lens. The focal length of the receiver (Rx) lens was designed so that the entire sub-FOV was imaged onto the input facet of the light pipe, which funneled light into the avalanche photo diode (APD).

### 2.2. Lidar Transmitter Optical Design

#### Pupil Matching between MEMS Mirror and DMD

Since a DMD was used as the front end of the transmitter, the beam size, steering angle, and beam position were matched between the 2D-MEMS mirror and the Tx-DMD while satisfying the condition that the beam footprint was smaller than each of the device areas. Figure 2 schematically depicts the optical train of the lidar transmitter. The light from the laser diode (LD) was collimated by lens L1, and the astigmatism was corrected by a cylindrical lens placed between L1 and L2 (LD focusing lens). The fine and coarse beam steering required the matching of the pupils, i.e., the image of the 2D-MEMS mirror was formed on the Tx-DMD via L3 (field lens) and L4 (collimator). In this manner, the beam stayed on the DMD while being expanded by a telescope with a magnification of $f_{4}/f_{2}$. The focal lengths of lenses L3 ($f_{3}$) and L4 ($f_{4}$) were determined by solving a system transfer matrix from the 2D-MEMS mirror to the DMD for the given distance from the 2D-MEMS mirror to L3 ($t_{1}$), L4, and the DMD ($t_{2}$) at a magnification of $m=\frac{D_{DMD}}{D_{MEMS}}$, and given by
(1)$$f_{3}=\frac{{\left(D_{DMD}\;t_{1}\right)}^{2}}{D_{DMD}^{2}\;t_{1}+D_{DMD}\;D_{MEMS}\;t_{1}-t_{2}D_{MEMS}^{2}}$$
(2)$$f_{4}=\frac{D_{DMD}\;t_{1}}{\;D_{MEMS}}$$
where $D_{DMD}$ and $D_{MEMS}$ are the beam sizes at the positions of the 2D-MEMS mirror and DMD, respectively. The focal lengths of L3 and L4 were identified as $f_{3}$ = 35.4 mm and $f_{4}=127.26$ mm, respectively, for the values $D_{DMD}$ = 7.07 mm, $D_{MEMS}$ = 2.5 mm, $t_{1}$ = 45 mm, and $t_{2}$ = 30 mm. The focal lengths of L1 and L2 were arbitrary as long as the beam hitting the 2D-MEMS mirror was not vignetted.

Figure 3 shows a ray-tracing diagram of the transmitter based on the formulation. A nanosecond laser from a laser diode (LS9-25-4-S10-00, Laser Components, Bedford, MA, USA) was collimated by an f = 4.51 mm L1 aspheric collimator (C230TMD-B, Thorlabs, Newton, MA, USA). An f = 100 mm cylindrical lens (#34-670, Edmund Optics, Barrington, IL, USA) was placed after an f = 50 mm L2 focusing lens (LA1213-B, Thorlabs, Newton, MA, USA) to correct for the astigmatism of the laser diode. A 2D-MEMS mirror (#S30262, Mirrorcle, Fremont, CA, USA) deflected the converging beam at 11.47(*H*) × 11.47(*V*) degrees and was imaged onto the transmitter DMD (DLP7000, Texas Instruments, Dallas, TX, USA) via a field lens (L3: LA1422-B, Thorlabs, Newton, MA, USA) and collimating lens (L4: #38-380, Edmund Optics, Barrington, IL, USA).

### 2.3. Lidar Receiver Optical Design

The second DMD was employed for the receiver to adaptively match the sub-FOVs of the transmitter to the sub-FOVs of the receiver. The receiver employed a light pipe to funnel photons within the sub-FOV into an avalanche photo diode (APD) (Figure 1a). The full angular extent of the sub-FOV of the receiver ($FOV_{sub,rec}$) was defined by the size of the input facet of the light pipe ($D_{lp}$) and the focal length of the receiver lens ($f_{rec}$), and is given by
(3)$$FOV_{sub,rec}=2ArcTan\left(\frac{D_{lp}}{2\;f_{rec}}\right)$$
provided that the angular extent of the light at the exit side of the light pipe did not exceed 180 degrees. The receiver accommodated $FOV_{sub,rec}=$ 10.98 degrees, with a 2× light pipe ($D_{lp}$ = 5 mm, 50 mm length) and $f_{rec}$ = 26 mm. With a lens of NA 0.55 and at 2× magnification, the extent of the output angle from the output facet of the light pipe was 66 degrees, which satisfies the condition that the ray extent of the exit side was smaller than 180 degrees.

Figure 4 shows the optical layout and design parameters of the lidar receiver. The reflected beam from the target was received by an Rx DMD (DLP7000, Texas Instruments, Dallas, TX, USA), which was synchronized with the Tx DMD (DLP7000, Texas Instruments, Dallas, TX, USA). A receiver lens (ACL3026U-B Thorlabs, Newton, MA, USA) imaged the object onto the edge of a 2x-concentration light pipe (#63-103, Edmund Optics, Barrington, IL, USA) at bandpass filters at central wavelength 905 nm and a bandwidth of 25 nm (FL905-25, Thorlabs, Newton, MA, USA). A silicon avalanche photo diode (APD130A, Thorlabs, Newton, MA, USA) was placed at the exit side of the light pipe to collect the photons. Figure 5 shows a photograph of the entire lidar system and the ray-tracing diagram.

### 2.4. Control Electronics

Figure 6 shows a block diagram of the control electronics. A microcontroller (Arduino Uno, Italy) sequenced and synchronized a pulsed laser diode, 2D-MEMS, Tx-, Rx-DMD, and the ToF measurement electronics. In the measurement sequence, first, the 2D-MEMS mirror was set to the starting position of the sub-FOV consisting of 16 by 12 points. Then the laser, Tx, and Rx DMDs were triggered to scan over seven diffraction orders for the 2D-MEMS position. For each of the laser pulses, the time-of-flight (ToF) detection was employed with the time-to-digital conversion (TDC7200, Texas Instruments, Dallas, TX, USA) circuit and in-house constant fractional discriminator (CFD) to correct for the amplitude-dependent signal walk that commonly occurs due to varying target distances [9]. The ToF values were then converted into distance values, and they were sorted by Matlab into a 12×112 array according to the scanning sequence steered by the 2D-MEMS mirror and DMD. This sequence was repeated at the next 2D-MEMS position until all (12 × 16) × 7 = 1344 points were scanned.

Figure 7 shows the long-exposure pattern of the lidar transmitter along with a pattern predicted by a ray-tracing simulation. The horizontal and vertical FOVs were at 38 and 7 degrees, respectively. The arc-like distortion of the beam pattern within a single diffraction order and across the diffraction orders were due to the 45-degree angle of incidence from the 2D-MEMS mirror to the DMD. The divergence of the spot was evaluated by measuring the extent of the beam at 1, 2, and 3 m (Figure 8). The beam divergence is shown in Table 1. Spot size and spot overlapping were measured at 1, 2, and 3 m from the transmitter DMD and are depicted in Figure 8a–c, respectively. The single spot had a divergence of 0.63(H)×0.172(V) degrees. The spot separation was 0.266(H)×0.7(V) degrees.

## 3. Experimental Results

The lidar system performance was measured by the diffraction efficiency of the DMD, the distance accuracy, lateral angular resolution, and distance resolution, and by capturing still and live images by following a procedure to test lidar systems found in Ref. [9].

### 3.1. Diffraction Efficiency

Table 2 shows the diffraction efficiency of the DMD measured for all the diffraction orders, −3 to +3. The signal for each diffraction order was measured with an APD (Thorlabs APD 130A, Thorlabs, NJ, USA) placed at 50 cm from the Tx DMD with a focusing lens to capture the diffracted beam from the Tx DMD, which funneled captured photons into the active area of the APD (effective area of detection is 1 mm in diameter). The diffracted power was normalized to the incident laser power to the Tx DMD, which was measured by the same configuration while replacing the Tx DMD with a mirror. The diffraction efficiency of the DLP7000 used for the lidar agrees with the diffraction efficiency of the DLP3000 reported in Ref. [17].

### 3.2. Distance Accuracy

Since the total FOV was divided into seven sub-FOVs corresponding to the diffraction orders of the DMD, the distance accuracies of sub-FOVs were individually evaluated. In the measurement, highly reflective white cardboard was used as a target and placed within the scanning range of each of the diffraction orders. The distance measurement was carried out by translating the target in 10 cm increments over a range of 90 cm to 500 cm, as shown in Figure 9. Across the seven diffraction orders (sub-FOVs), the system exhibited good linearity as well as consistency in distance accuracy.

### 3.3. Distance Resolution

The distance resolution was tested by placing cardboard at 100 cm and 101 cm away from the Tx DMD while covering all the scanning points of 12 × 16 within the sub-FOV of each of the diffraction orders. The 12 × 16 distance values were plotted as a histogram to show their distribution, and a normal distribution curve fit was added (Figure 10). It is shown in Figure 10 that the peaks of the curves discernible for separation are as small as 1 cm.

### 3.4. Angular Resolution

The horizontal and vertical angular resolutions were tested with two white cardboards, placed 1 m away within the scanning extent of a single diffraction order, creating a slit, as shown in Figure 11a,b. The angular resolution was tested for each of the diffraction orders (sub-FOVs) by narrowing down the width of the slit until the gap shown in the lidar image was no longer resolvable. The slit width was measured with a ruler, and the corresponding angle subtended by the slit was calculated. This process was repeated for all diffraction orders. The horizontal and vertical angular resolutions were experimentally identified as 0.57°(H) and 0.23° (V), respectively.

### 3.5. Still Lidar Image Capture

Static objects were captured to test the system’s ability to measure the distance accuracy and detect the object shape. As a static object, a diagonal cutout of cardboard was placed at the −3rd and 3rd orders, and the letters “L”, “I”, “D”, “A”, and “R” were placed at the −2nd, −1st, 0th, 1st, and 2nd orders, respectively, as shown in Figure 12. The actual distances of the targets of “L”, “I”, “D”, “A”, and “R” were 70 cm, 110 cm, 95 cm, 110 cm, and 70 cm from left to right. The color on the live image visually represents the distances of the targets; the distance values are shown by the color scale on the right. The measured mean distance value for each diffraction order is given at the top of the diagram and agrees with the actual distance of each letter.

### 3.6. Live Lidar Image Capture

A piece of white cardboard placed 50 cm away was translated in the lateral direction across the entire FOV of the lidar system. The corresponding video footage of the live image shows that the reconstruction of the moving object was accurately depicted, as shown in Figure 13a. Since the target’s axial distance from the DMD was fixed at about 90 cm, the distance represented by the color scale was unchanged. As the target was translated from left to right, the live image reconstruction shows the corresponding target’s change of location. To demonstrate the detection of motion of multiple targets (toy cars) in the axial direction, the first toy car was driven away from the lidar, and the second toy car was driven towards the lidar, while their lateral location was fixed at the −1st order and 2nd order, respectively. As shown in Figure 13, as the first car drove away, the color changed from blue (close distance) to yellow (far distance). Conversely, as the second car drove toward the system, the color changed from yellow to blue. The frame rate for the lidar image capture was 0.588 fps.

## 4. Discussion

### 4.1. Distortion in Beam Pattern

As the beam scan pattern depicted in Figure 7a shows, the pattern is symmetrical but suffers from an arc-like artifact due to the out-of-plane incidence of the beam from the 2D-MEMS mirror to the DMD (Figure 1a). The arc-like scan might be beneficial for some lidar imaging scenarios, for example, capturing the lidar image of a car in front of the lidar at the center of the road, along with buildings along the road. Alternatively, the in-plane illumination geometry depicted in Figure 14 provides the rectangular scan geometry while slight distortion of the pattern is observed in the scan pattern, in particular for the large diffraction orders. This can be understood as the angle of diffraction has inherent non-linearity in the grating equation, particularly at higher diffraction orders.

### 4.2. Angular Resolution

From the angular resolution test (Figure 11), the resolution of the system is identified as 0.57°×0.23° (H×V). As is shown in Figure 7b and Figure 8, the horizontal spots are partially overlapped, while the vertical spots are spatially separated. Figure 15 schematically depicts the geometry of the slit and the degree of overlapping of the spots in the horizontal direction. As seen in Figure 15a,c, the minimum detectable slit width $\Delta W_{H}$, $\Delta W_{V}$, is not affected by the angular separation of the spots in the horizontal and vertical directions, $\Delta S_{H},\;\Delta S_{V},$ but is solely determined by the angular extent of the spots. For example, Figure 15a,b shows the configurations for the oversampled and normal sampled cases, respectively. It is obvious that regardless of the angular sampling frequency, the resolvable angular separation for both cases is equal to $\Delta W_{H}$. Although decreasing the angular separation of the spot does not yield a better angular resolution, it provides a higher sampling density and, therefore, enables higher fidelity in lidar image reconstruction.

As the calculated and measured angular resolution in Table 1 shows, the horizontal angular resolution is limited by $W_{H}$. The $W_{H}$ is determined by the conservation of etendue between the two locations in the optical train—the active layer of the laser diode (0.085(H)×0.01(V)mm, NA=0.13(H)×0.37(V)) and at DMD (10(H)×10(V)mm, NA=0.087)—while satisfying the condition such that the beam footprint on the 2D-MEMS mirror is smaller than the MEMS mirror area, 3.6 mm in diameter. Since the etendue of the DMD is much larger than that of the laser diode (LD), all the power from the LD is transferred to the DMD. In addition, the astigmatism of the LD is corrected by the cylindrical lens, and the horizontal elongation of the spot is due to the wider geometrical extent of the active layer of the LD in the horizontal direction. The horizontal angular resolution can be increased by decreasing the horizontal width of the active layer of the laser; however, the energy per pulse decreases accordingly at the expense of a higher horizontal resolution.

### 4.3. Frame Rate

Since the demonstrated lidar system employed a single-point detector, the scan speed was limited by the time required for the ToF measurement and/or the beam scanning speed. Currently, the scanning speed is 790 pts/s or 0.588 fps for (12 × 16) × 7 = 1344 points, which is limited by the data transfer of the serial monitor from the microcontroller to the live image display. Without the data transfer limitation, the maximum scan speed is primarily limited by the frame rate of the DMD, which is currently 23 kHz. Diffractive beam steering requires resetting all the mirrors to the off-state prior to the next diffractive steering. This reset halves the scan speed to 11.5 kHz. Alternatively, diffractive steering can be incorporated by using the bi-directional transition from all on-state to all-off-state to fully take advantage of the frame rate of the DMD. Further increases in the scanning point range require a two-dimensional arrayed detector, such as a single photon avalanche diode (SPAD) array and a multi-pixel photon counting (MPPC) array. The proposed quasi-solid-state-lidar architecture forms a two-dimensional lidar image by the point-by-point scanning of the lasers by hybrid optical architecture employing a 2D-MEMS mirror and a DMD. With the 2D array detector, an even faster scan rate is expected, without altering the Tx optical architecture. For example, an MPPC array of 32 × 32 pixels was reported [20], with a 1 kHz data acquisition rate. That would achieve a 1 M points/s data acquisition rate while still employing the proposed optical architecture, synchronizing the Tx and Rx DMD to expand the FOV.

## 5. Conclusions

We have demonstrated a quasi-solid-state lidar system employing two kinds of MEMS systems—a two-dimensional MEMS mirror and a Texas Instruments digital micromirror device (DMD). The hybrid MEMS lidar system provides a way for beam steering that achieves a large-scan angle, fast-scan rate, and high resolution utilizing commercially available DMD and a micro electro mechanical system (MEMS) based on a two-dimensional scan mirror while maintaining implementation as a quasi-solid-state lidar. The coarse scanning from diffractive beam steering by the DMD provides a large scan angle and fast scan rate. Beam steering by the two-dimensional MEMS-based mirror increases the number of scanning points, which yields a high resolution for object detection. By employing the DMD as a receiver, its large aperture area allows the lidar system to detect a long range since the maximum range scales with the square root of the aperture area. The frame rate to scan 12 × 112 points over 37 degrees is currently limited to 0.588 fps by the data transfer rate over a serial data channel. Aside from the data transfer limitation, a further increase in the data rate is expected by replacing the single-point detector with an arrayed detector, such as a 32 × 32-pixel multi-pixel photon counting (MPPC) array, while not altering the current optical architecture of the lidar transmitter two-dimensional MEMS and DMD.

## Figures and Tables

**Figure 1 micromachines-13-01444-f001:**
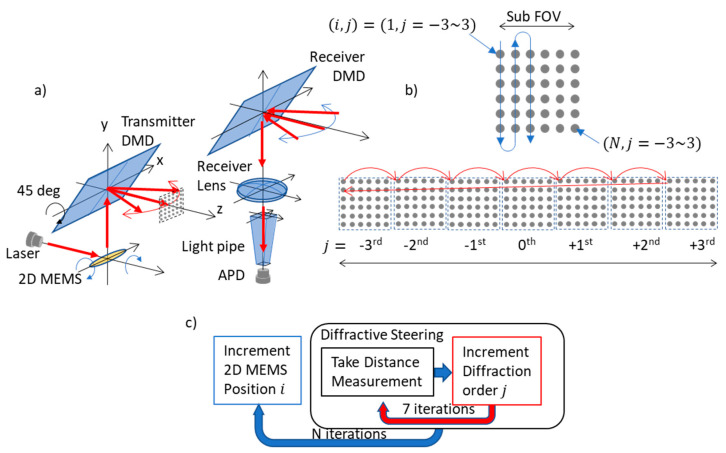
(**a**) Hybrid lidar optical architecture employing 2D-MEMS mirror, transmitter (Tx) DMD, and receiver (Rx) DMD. (**b**) 2D-MEMS mirror scan of the sub-FOV of N scanning points. (**c**) The DMD sequentially steers the scanning point within the sub-FOV into seven diffraction orders in an interleaved manner.

**Figure 2 micromachines-13-01444-f002:**
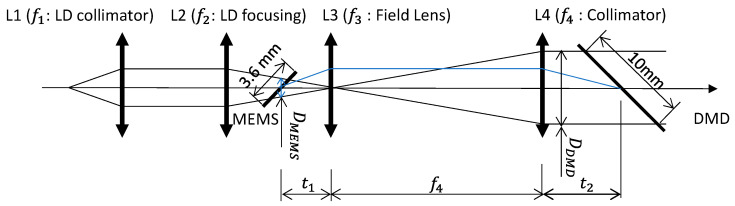
A first-order diagram of the lidar transmitter to expand the beam while matching the pupils of the 2D-MEMS mirror and DMD.

**Figure 3 micromachines-13-01444-f003:**
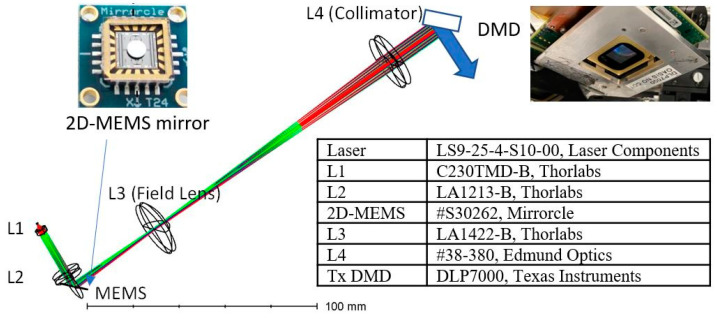
A ray-tracing diagram of the lidar transmitter to expand the beam while matching the pupils of the 2D-MEMS mirror and DMD.

**Figure 4 micromachines-13-01444-f004:**
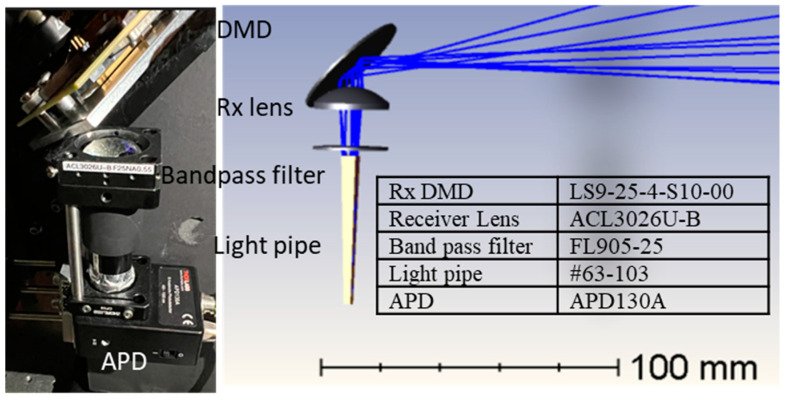
Photograph (**left**) and a ray-tracing diagram of the lidar receiver (**right**).

**Figure 5 micromachines-13-01444-f005:**
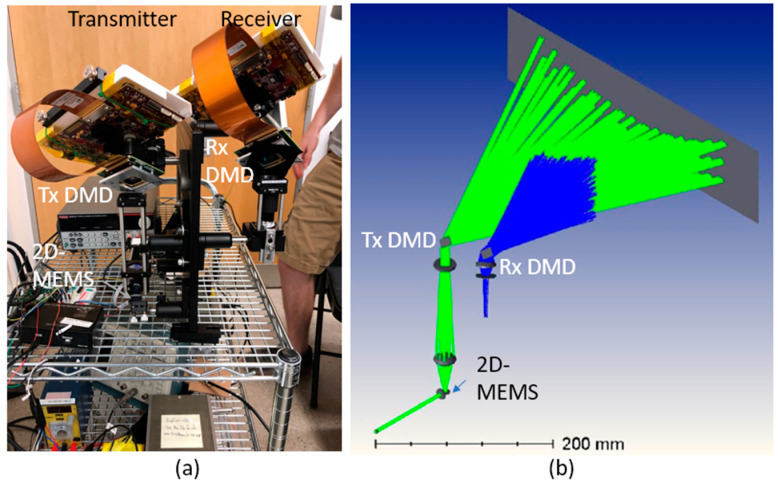
(**a**) Photograph and (**b**) a ray-tracing diagram of the lidar transmitter and receiver.

**Figure 6 micromachines-13-01444-f006:**
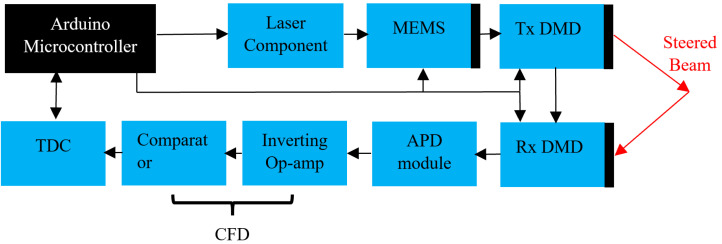
Block diagram of the system operation with time-of-flight (ToF) circuitry. The microcontroller triggered the laser, Tx- and Rx-DMD, and 2D-MEMS mirror. For each of the laser pulses, the ToF measurement was employed.

**Figure 7 micromachines-13-01444-f007:**
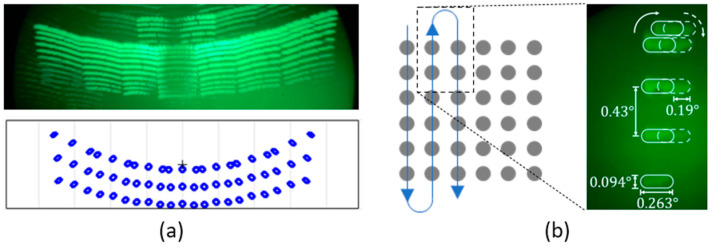
(**a**) A long exposure of the scanning pattern captured by an IR viewer (top), and simulated scanning pattern (bottom). (**b**) Beam scanning pattern at the top edge of the sub-FOV with beam divergence values. The “+” indicates the center of 0 th order sub-FOV.

**Figure 8 micromachines-13-01444-f008:**
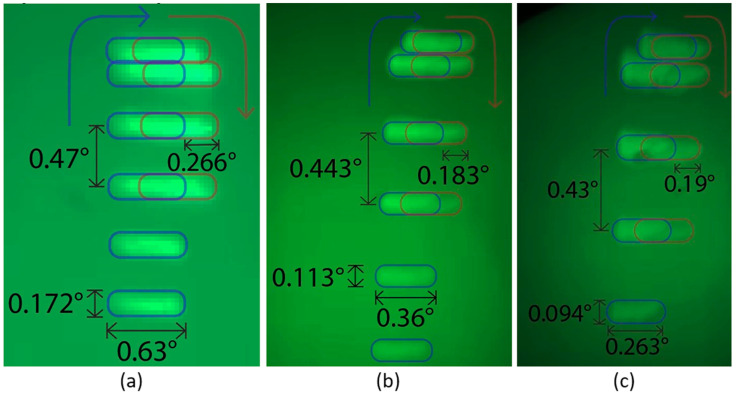
Angular spot separation and spot extent for horizontal and vertical directions for (**a**) 1 m, (**b**) 2 m, and (**c**) 3 m distances. The pictures show the top portion of the sub-FOV depicted in Figure 8b.

**Figure 9 micromachines-13-01444-f009:**
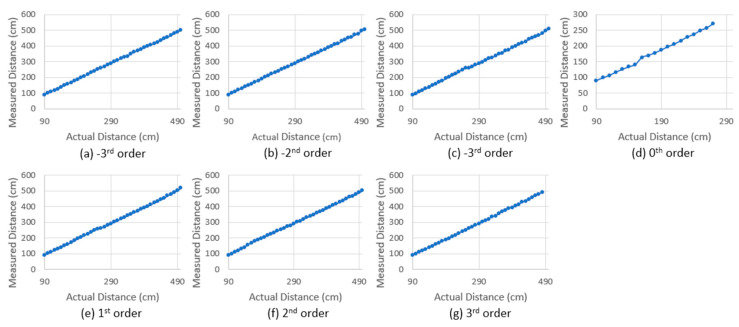
Measured distance vs actual distance for (**a**) −3rd, (**b**) −2nd, (**c**) −1st, (**d**) 0th, (**e**) +1st, (**f**) +2nd, and (**g**) +3rd orders (seven diffraction orders). The measurements show good and consistent linearity across the seven sub-FOVs, corresponding to the −3 to +3 diffraction orders of the Tx- and Rx-DMDs.

**Figure 10 micromachines-13-01444-f010:**
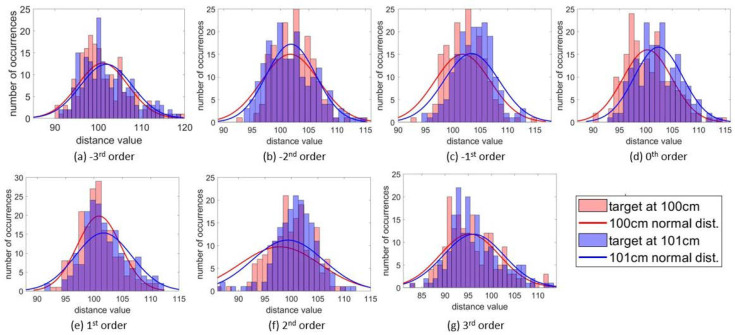
Histogram of distance values for (**a**) −3rd, (**b**) −2nd, (**c**) −1st, (**d**) 0th, (**e**) +1st, (**f**) +2nd, and (**g**) +3rd orders. Across the 7 sub-FOVs, the distance accuracy was at least 1 cm.

**Figure 11 micromachines-13-01444-f011:**
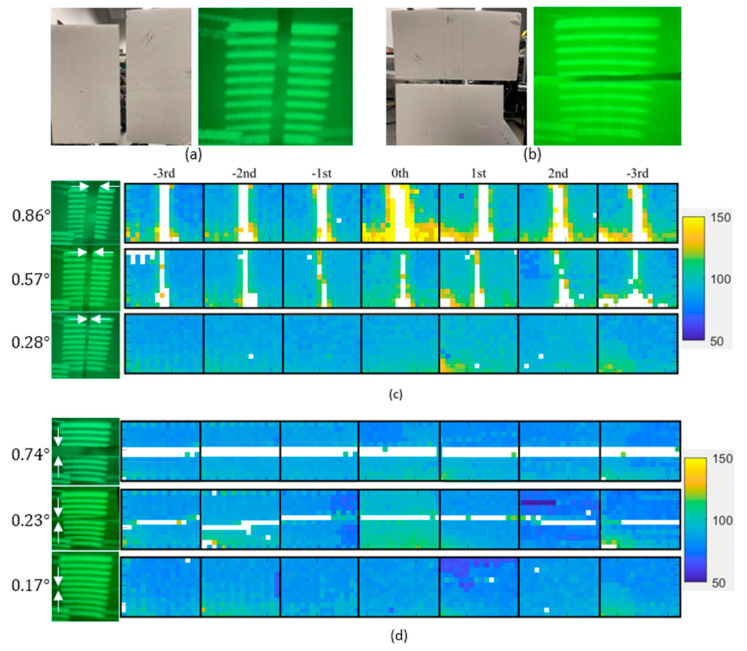
(**a**) Vertical slit placed at a single diffraction order and corresponding beam pattern on the target. (**b**) Horizontal slit placed at a single diffraction order and corresponding beam pattern on the target. (**c**) Lidar image corresponding to the vertical slit target depicted in (**a**). (**d**) Lidar image corresponding to the horizontal slit target depicted in (**b**).

**Figure 12 micromachines-13-01444-f012:**
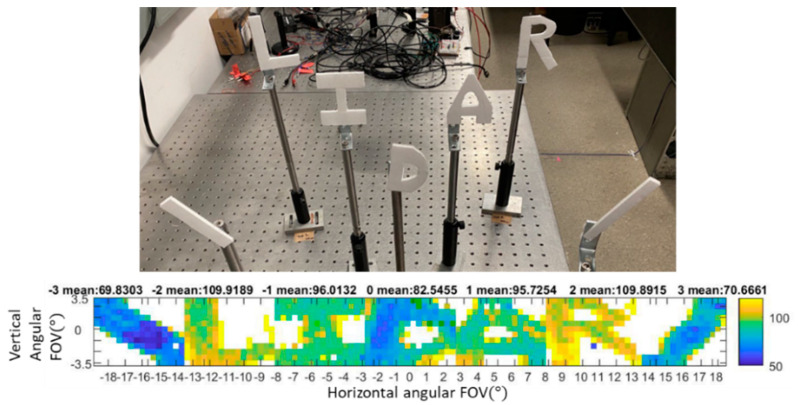
Target placement of “LIDAR” object and its corresponding lidar image reconstruction. The measured distance values indicated above the figure show good agreement with the actual distances of the targets.

**Figure 13 micromachines-13-01444-f013:**
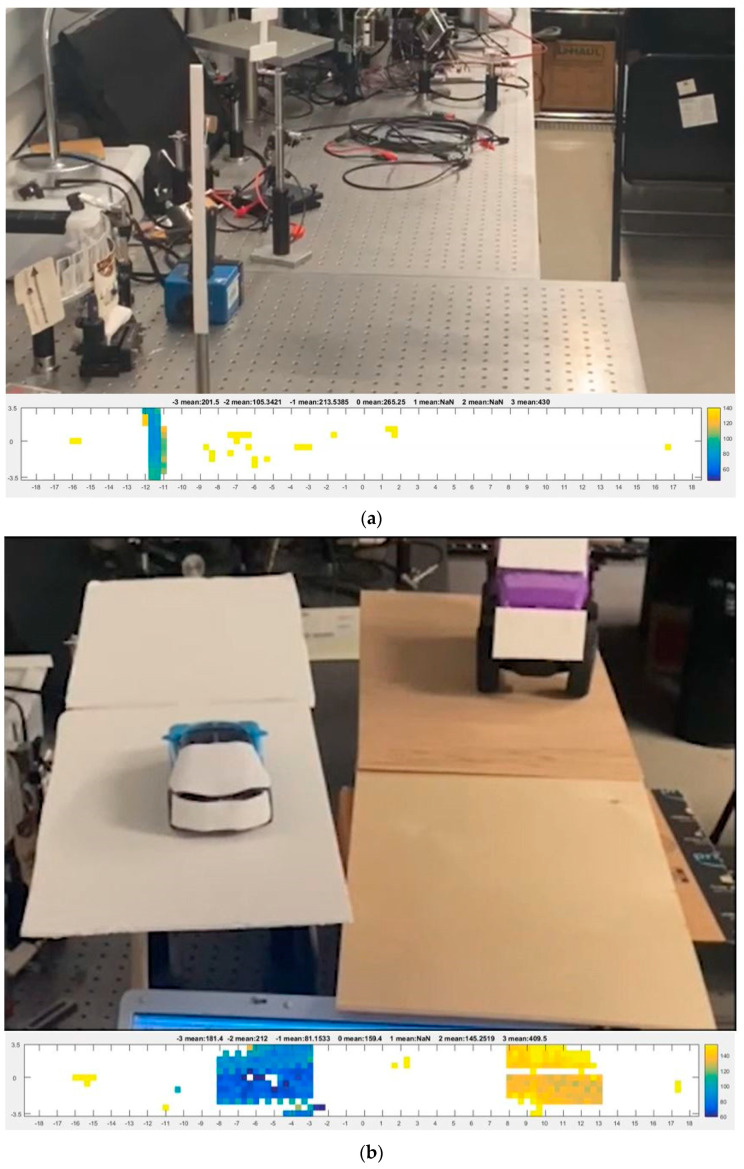
Video footage of (**a**) lateral translation of a bar-like target and its corresponding lidar image reconstruction; see Appendix A. (**b**) Axial translation of two toy cars moving in opposite directions and the corresponding lidar image reconstruction; see Appendix A.

**Figure 14 micromachines-13-01444-f014:**
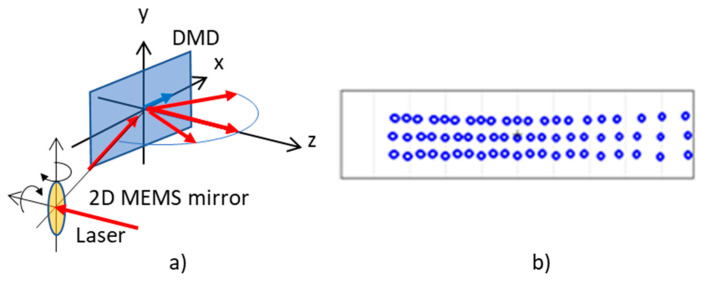
(**a**) Arrangement of the 2D-MEMS mirror and DMD as in-plane illumination and (**b**) corresponding beam pattern over 7 sub-FOVs. The in-plane geometry eliminates the arc-like scanning pattern found in the out-of-the-plane geometry. Slight distortion of the scanning pattern is observed at the edge of the FOV due to the non-linearity in the grating equation.

**Figure 15 micromachines-13-01444-f015:**
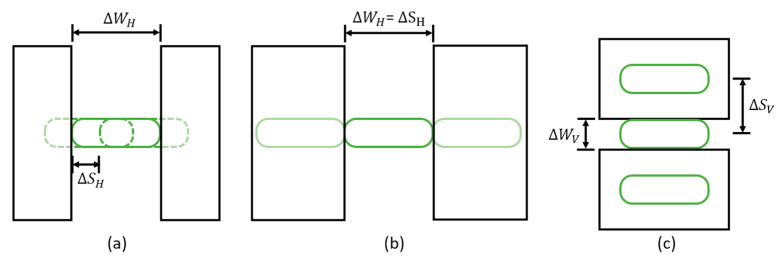
Configuration for the minimum resolvable angular separation for (**a**) the oversampled and (**b**) normal sampled cases in the horizontal direction and (**c**) the vertical direction.

**Table 1 micromachines-13-01444-t001:** Angular spot size and angular spot separation for horizontal and vertical directions.

	Angular Spot Size (°)		Angular Spot Separation (°)	
Distance (m)	Horizontal	Vertical	Horizontal	Vertical
1	0.629	0.172	0.2	0.472
2	0.36	0.113	0.183	0.443
3	0.263	0.095	0.19	0.429

**Table 2 micromachines-13-01444-t002:** Diffraction efficiency of the DMD.

Diffraction Order	−3	−2	−1	0	1	2	3
Efficiency	0.4	0.56	0.5	0.74	0.52	0.48	0.34

## Data Availability

Data is available upon request for the corresponding author.

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
