# Peer review of "All-MEMS Lidar Using Hybrid Optical Architecture with Digital Micromirror Devices and a 2D-MEMS Mirror"

_micromachines, 2022, doi:10.3390/mi13091444_

Round 1

Reviewer 1 Report

This manuscript demonstrates a MEMS lidar system which employs two kinds of MEMS devices: a DMD and 2-dimensional MEMS mirror for transmitter, and a DMD with light-pipe-based optics for receiver. The optical structure of this hybrid MEMS lidar was present, followed by which are the detailed experimental results. The MEMS lidar system in this paper has good scalability, which is an interesting topic in future direction. Only some small issues need to be addressed accordingly before publication.

1. Please correct some writing mistakes, e.g. Figure 8. “Angular” instead of “angular” in line 250.

2. Please provide the full names of some abbreviations when they appear first in the text. e.g. SPAD and MPPC in line 403.

Reviewer 2 Report

This paper demonstrates a lidar system employing two kind of MEMS system, 2-dimensionaol MEMS mirror and DMD. The paper could be published after minor revised.

1)   Some expressions in the abstract are not appropriate, for example “……area of 140mm2. In the receiver, a 2nd Digital Micromirror”.

2)   Figure 3 and 13 are not very clear and strict.

3)   The lidar in this paper is not all-solid-state lidar.
